# Development and validation of an interpretable longitudinal preeclampsia risk prediction using machine learning

**Braden W. Eberhard[1], Raphael Y. Cohen[1,2], Nolan Wheeler[1], Ricardo Kleinlein[1], John Rigoni[1], David W. Bates[3,4], Kathryn J. Gray[5]☸*, Vesela P. Kovacheva[1]☸***

**1** Department of Anesthesiology, Perioperative and Pain Medicine, Brigham and Women's Hospital, Harvard Medical School, Boston, Massachusetts, United States of America, **2** Health Data Analytics Institute, Dedham, Massachusetts, United States of America, **3** Division of General Internal Medicine and Primary Care, Brigham and Women's Hospital, Boston, Massachusetts, United States of America, **4** Department of Health Care Policy and Management, Harvard T. H. Chan School of Public Health, Boston, Massachusetts, United States of America, **5** Department of Obstetrics and Gynecology, University of Washington, Seattle, Washington, United States of America

☸ These authors contributed equally to this work.
* kgray5@uw.edu (KJG); vkovacheva@bwh.harvard.edu (VPK).

## Abstract

Preeclampsia is a pregnancy-specific disease characterized by new onset hypertension after 20 weeks of gestation that affects 2–8% of all pregnancies and contributes to up to 26% of maternal deaths. Despite extensive clinical research, current predictive tools fail to identify up to 66% of patients who develop preeclampsia. We sought to develop a tool to longitudinally predict preeclampsia risk. In this retrospective model development and validation study, we examined a large cohort of patients who delivered at three hospitals in the New England region between 05/2015 and 05/2023. We used sociodemographic, clinical diagnoses, family history, laboratory, and vital signs data. For external validation, we used the Nulliparous Pregnancy Outcomes Study: Monitoring Mothers-to-Be (nuMoM2b) cohort (2010–2013), which contained data from eight external sites in the US. Models were developed at eight gestational time points using logistic regression, elastic net, naïve-Bayes, random forest, xgboost, and deep neural network methods. We used Shapley values to investigate the relationships between features. Our study population (N = 101,357) had an incidence of preeclampsia of 6.1% (N = 6,160). Model AUCs ranged from 0.71–0.80 (95%CI 0.69–0.82), externally validated in the nuMoM2b cohort with an AUC range of 0.57–0.70 (95%CI 0.55–0.73). No significant differences in performance were found based on race and ethnicity. As these novel models identify more patients at risk for developing preeclampsia, the benefits of this approach need to be balanced with the need for surveillance in a larger at-risk population. This novel preeclampsia prediction approach allows clinicians to identify at-risk patients early and provide personalized predictions throughout pregnancy.

**Data availability statement:** Data Availability Statement. The data used in this study are from Mass General Brigham patients and from the Nulliparous Pregnancy Outcomes Study: Monitoring Mothers-to-Be (nuMoM2b) cohort. The data from Mass General Brigham patients cannot be shared publicly because of restrictions related to patient privacy and confidentiality. Data are available from the Mass General Brigham Institutional Review Board (contact via email, IRB@mgb.org, phone +1-857-282-1900, address 75 Francis St, Boston, MA, USA) for researchers who meet the criteria for access to confidential data. The Nulliparous Pregnancy Outcomes Study: Monitoring Mothers-to-Be (nuMoM2b) cohort is publicly available from NICHD DASH Data and Specimen Hub. We share all code relevant to this work at https:// github.com/KovachevaLab.

**Funding:** Funding. KJG reports funding from NIH/NHLBI (https://www.nhlbi.nih.gov, grants K08 HL146963, K08 HL146963-02S1, and R03 HL162756). VPK reports funding from the NIH/NHLBI (https://www.nhlbi.nih.gov, grant 1K08HL161326), APSF (https://www.apsf.org, grant IIR) and BWH (https://brighamandwom-ens.org, Ignite, Mary Horrigan Connors Center). The funders played no role in the study design

**Competing interests:** Competing interests. KJG has served as a consultant to Illumina Inc., Aetion, Roche, and BillionToOne outside the scope of the submitted work. DWB reports grants and personal fees from EarlySense, personal fees from CDI Negev, equity from Valera Health, equity from CLEW, equity from MDClone, personal fees and equity from AESOP Technology, personal fees and equity from FeelBetter, and grants from IBM Watson Health, outside the submitted work. VPK reports consulting fees from Avania CRO unrelated to the current work. VKP and RYC report patent #WO2021119593A1 for the control of a therapeutic delivery system assigned to Mass General Brigham.

## Introduction

Preeclampsia is a pregnancy-specific disorder characterized by new onset hypertension and proteinuria after 20 weeks gestation [1] that complicates 2–8% of pregnancies [2,3]. Worldwide, preeclampsia accounts for up to 10 million cases annually and is responsible for approximately 26% of maternal deaths in low- and middle-income countries [4,5]. In addition to elevated systolic (>140 mmHg) or diastolic (>90 mmHg) blood pressure, patients can develop progressive end-organ damage characterized by proteinuria, elevated liver enzymes, pulmonary edema, seizures, and death. The only definitive treatment is delivery, and as a result, preeclampsia is the leading cause of iatrogenic preterm birth with associated neonatal morbidity and mortality.

Despite extensive research, the upstream causative factors for preeclampsia remain unknown. It is hypothesized that abnormal placental development may be a large contributor [6]. Additional potential factors influencing preeclampsia include genetic predisposition, maternal immunologic and inflammatory responses, environmental factors such as air pollution and dietary intake, and socioeconomic determinants including limited healthcare access and chronic stress [1,4,5]. Current clinical practice is focused on identifying patients at risk based on established clinical criteria, which include demographic characteristics and medical history [1]. Maternal risk can be mitigated by close surveillance using frequent home blood pressure monitoring and prophylaxis with low-dose aspirin based on the presence of at least one high-risk or two moderate-risk factors [7]. However, more than half of pregnancies, in particular those in nulliparous individuals, end with preeclampsia in the absence of risk factors [8,9]. In addition, little is known about the trajectory of risk and the rate of change in risk across pregnancy.

As tools to predict preeclampsia are lacking, multiple predictive models have been developed. Most models utilize clinical risk factors combined with biomarkers, which may not be available in routine care [8,10]. The majority of these models are also derived from well-curated datasets using statistical methods [8–11]. Recently, a few accurate machine learning models have been developed [11]; however, the full potential of electronic health record (EHR) data has not been realized. Models developed to date have shown promise using small, constrained data sets but have been lacking in predictive power, accessibility, and generalizability. Machine learning models based on the entire EHR have great potential and outperform a rule-based algorithm that relies on established clinical criteria [12,13].

Using the readily available rich data source of the EHR and machine learning to integrate data with complex relationships over time, we sought to develop accurate predictions of the risk for preeclampsia throughout pregnancy. We created a large patient cohort using longitudinal data from three hospitals from a large healthcare system in the Northeastern US. We developed a system of multiple machine-learning models throughout pregnancy and investigated their performance in preeclampsia prediction. Implementing this type of multi-model system to automatically operate within the EHR by ingesting data as it becomes available and generating predictions longitudinally during pregnancy would allow physicians to timely identify, triage, and modify surveillance and levels of care for the highest-risk patients.

Given the technical complexity of machine learning terminology, we have included definitions and clinical relevance of key terms (Table 1) to facilitate reader comprehension.

## Related works

Existing clinical approaches to preeclampsia risk prediction have significant limitations, often missing up to 66% of cases due to reliance on limited clinical risk factors, such as prior history, chronic hypertension, or diabetes alone [8,9]. Over the past decade, multiple predictive models (Table 2) have been developed using varied data sources, ranging from basic clinical factors and vital signs to advanced biomarkers, genetics, and ultrasound findings [8–14].

Early statistical models using logistic regression (e.g., the Fetal Medicine Foundation's first-trimester combined screening test) incorporated clinical, biochemical (Pregnancy-associated Plasma Protein-A, PAPP-A, placental growth factor, PlGF), and biophysical markers (mean arterial pressure, uterine artery Doppler) to achieve high accuracy (AUC ~ 0.90) for early-onset preeclampsia [10,15]. However, these require specialized tests not routinely available, limiting broader applicability.

Biomarker-based approaches, particularly the soluble fms-like tyrosine kinase-1/placental growth factor (sFlt-1/PlGF) ratio, demonstrate strong predictive performance (high negative predictive value) in symptomatic women later in pregnancy and are used clinically in some European guidelines [16]. Still, their utility for general early-pregnancy screening remains limited due to timing and resource constraints.

More recently, machine learning methods, such as random forests, gradient boosting (xgboost), and neural networks, have shown potential by capturing complex nonlinear relationships inherent in clinical data [13,15,17,18]. For example,

**Table 1. Explanation of machine learning and statistical terms.**

| Term | Definition and Clinical Relevance |
|---|---|
| Machine Learning (ML) | A subset of artificial intelligence techniques that allows algorithms to learn patterns from data without explicit rules, improving predictive accuracy. |
| Area Under the Curve (AUC) | A statistical measure of how well a prediction model distinguishes between two classes (e.g., presence or absence of preeclampsia). AUC values range from 0.5 (no predictive power) to 1.0 (perfect prediction). |
| Logistic Regression | A statistical method used to predict the probability of an outcome based on one or more predictor variables. Suitable for binary outcomes (e.g., presence or absence of disease). |
| Elastic Net | A regression model combining regularization techniques to manage numerous predictors, effectively reducing model complexity and improving prediction stability. |
| Naive Bayes | A classification algorithm based on Bayes' theorem that assumes independence between predictor variables. Despite its simplicity, it performs well with large datasets. |
| Random Forest | A machine learning algorithm using multiple decision trees to improve predictive accuracy and control overfitting by combining individual predictions. |
| Xgboost (Gradient Boosting) | An advanced machine learning technique using sequential decision trees, focusing on reducing prediction errors of previous trees, providing high accuracy and efficiency with structured data. |
| Neural Network (Deep Learning) | Computational models inspired by human neural structures, capable of identifying complex, nonlinear relationships between variables through multiple layers of interconnected nodes ('neurons'). |
| SHAP (Shapley) Values | A technique providing interpretability of machine learning predictions by quantifying the impact of each feature on individual and overall predictions, based on cooperative game theory. |
| Data Leakage | A methodological error where information from outside the training set is inadvertently used during model training, leading to falsely inflated performance. |
| Oversampling (RandomOversampler) | A data balancing technique used to handle imbalanced datasets by randomly duplicating examples from the minority class, thus improving the model's ability to detect rare conditions (e.g., preeclampsia). |
| Internal vs. External Validation | Internal validation refers to evaluating a model's performance on a separate subset of data from the same source used for training, while external validation tests performance on an independent dataset from a different source or population. |

models by Jhee et al. and Marić et al. using gradient boosting and elastic net methods achieved AUCs ranging from 0.79–0.92 across various datasets and gestational ages, typically outperforming simpler logistic models [13,18]. Yet, deep learning methods have not consistently demonstrated superiority over ensemble models when applied to structured prenatal data [19].

Despite promising accuracy, most machine learning models have limitations, including insufficient external validation, reliance on non-routine biomarkers, and lack of interpretability, limiting clinical adoption [17,19]. For instance, Li et al. employed longitudinal EHR-based machine learning models with promising internal results but lacked external validation [12]. Specialized telehealth architectures, such as ILITIA by Moreira et al., utilize rule-based systems that simplify remote screening but do not fully leverage the complex relationships captured by machine learning [20].

Our study addresses these gaps by developing interpretable, longitudinal machine learning models using routinely collected clinical data from electronic health records, tested rigorously through internal and external validation. Unlike previous works, our approach uniquely explores temporal dynamics of predictors, uncovering novel indicators such as fluctuations in red blood cell counts, and utilizes Shapley values for enhanced interpretability and clinical relevance.

Ultimately, embedding such interpretable, machine-learning-driven predictive tools directly into routine obstetric care could transform the clinical management of preeclampsia, allowing personalized interventions and targeted surveillance.

## Methods

### Study population

This study was approved by the Mass General Brigham Institutional Review Board, protocol # 2020P002859, with a waiver of patient consent. We leveraged EHR data from three hospitals in the New England region with deliveries between May 2015 (when our institution implemented single-vendor EHR across all outpatient offices and inpatient sites) and May 2023. The data were pulled for research purposes between May 1, 2023, and June 2, 2023. The authors had access to information that could identify individual participants during or after data collection. Deliveries within the Mass General Brigham system were selected based on documentation of pregnancy greater than 20 weeks gestation and associated billing codes for cesarean or vaginal delivery [21]. The final dataset included data from patients for whom longitudinal data starting from the first prenatal visit before 14 weeks of gestation was available, and we analyzed each pregnancy

**Table 2. Performance and description of models to predict preeclampsia risk.**

| Study | Model Type | AUC | Patient Population | Timing | Key Predictors |
|---|---|---|---|---|---|
| Wright et al. (2019) [10] | Logistic regression | 0.80-0.95 | 61,174 | First Trimester | Clinical, Doppler, PlGF, PAPP-A |
| Sandstrom et al (2021) [11] | Shared-effects joint longitudinal models | 0.73-0.87 | 58,899 nulliparous | Longitudinal | Clinical |
| Li et al. (2022) [12] | Xgboost | 0.66-0.92 | 108,557 | Longitudinal | Clinical |
| Marić et al. (2020)[13] | Elastic Net, Gradient Boosting | 0.79–0.89 | 16,370 | First Trimester | Clinical |
| Kovacheva et al. (2024) [14] | Xgboost and Polygenic Risk Score | 0.74 | 1,125 pregnancies | First & Second Trimester | Genetic, Clinical |
| Espinoza et al. (2024) [16] | Biomarker (sFlt-1/PlGF Ratio) | 0.91 | 543 | Mid-to-Late Pregnancy | sFlt-1/PlGF Ratio |
| Ranjbar et al. (2024) [17] | Random Forest, Xgboost, Elastic Net | 0.86-0.97 | Systematic Review | First/Second Trimester | Clinical, Biomarkers |
| Jhee et al. (2019) [18] | Gradient Boosting | 0.92 | 11,006 | Third Trimester | Clinical |
| Liu et al. (2022) [19] | Neural Network, Random Forest | 0.86 | 11,057 | Early Pregnancy | Clinical, Doppler, PAPP-A |

Abbreviations: sFlt-1/PlGF, soluble fms-like tyrosine kinase-1/placental growth factor; PAPP-A, Pregnancy-associated Plasma Protein-A

independently. The results were externally validated using the Nulliparous Pregnancy Outcomes Study: Monitoring Mothers-to-Be (nuMoM2b) cohort [22]. These data were collected during 2010–2013 from eight external clinical sites across the USA with the goal of investigating adverse pregnancy outcomes in nulliparous individuals with singleton pregnancies. The detailed methodology steps are shown in Fig 1.

## Data processing

All data were ingested and processed using our machine learning platform, Medical Record Longitudinal Information AI System (MERLIN) [23], which extracts, transforms, and harmonizes data from the EHR. We censored our datasets at 14, 20, 24, 28, 32, 34, 36, and 38 weeks of gestation. We selected these time points based on the timing of routinely scheduled outpatient visits when new laboratory and vital signs data are obtained. For each of the eight datasets, we verified by inspecting the time stamps that only data obtained up to the selected time point were included to minimize the risk of data leakage. For every time point, patients were excluded if they had a date of delivery or date of diagnosis (if preeclampsia) prior to the weeks of gestation for that dataset.

## Model features and preeclampsia phenotype definition

All demographic, family history, medical history, laboratory studies, medications, insurance, vital signs, and procedural information data were extracted. We selected established risk factors associated with preeclampsia risk in published studies and guidelines [9,11–13,24,25]. We employed a median imputation strategy to handle missing data, and the features missing over 95% of entries were removed. As preeclampsia International Classification of Diseases (ICD) codes alone have limitations[4] and may result in data leakage, we defined the preeclampsia phenotype using a combination of laboratory values, blood pressure measurements, and ICD codes based on the established guidelines [24].

International Classification of Diseases (ICD) codes have well-documented limitations, including overall low accuracy in identifying preeclampsia. We found that only 69.1% of the patients with ICD 9 codes 642.4,642.5,642.6, and 642.7 and ICD 10 codes O11, O14, and O15 have preeclampsia, and commonly, those ICD codes may be assigned when the patients are admitted or evaluated for suspicion of preeclampsia, regardless of whether the diagnosis is later confirmed. To improve relevance to clinical practice, we developed a combination of laboratory values, blood pressure measurements, and ICD codes based on the established guidelines to define the preeclampsia phenotype in our datasets.

We randomly selected 400 cases based on ICD diagnoses for gestational hypertension, chronic hypertension, and preeclampsia. From these, 63 patients did not have hypertensive conditions, and the remaining 337 cases were used to develop the optimal combination of vital signs for our algorithm. The patient records were reviewed by clinical experts (V.K., K.G.), confirming that 126 patients had preeclampsia. The final algorithm, which was optimized for the number of blood pressure readings, blood pressure cutoffs, and ICD codes, demonstrated the best performance.

To address potential data leakage and result in inflation, we accounted for the fact that ICD codes are sometimes assigned after a patient has met laboratory and blood pressure criteria. Therefore, we used the time when the first blood pressure reading, or proteinuria, was reported as the time of preeclampsia diagnosis. The criteria for the final algorithm are as follows:

- Two blood pressure readings > 140/90 mmHg, 4 hours apart, combined with proteinuria.

- Repeated blood pressure in the severe range (7 readings of >160/100 mmHg in 2 hours)

- HELLP syndrome-related ICD diagnosis (ICD 9: 642.7, ICD 10: O14.1)

- Presence of eclampsia-related ICD diagnosis (ICD 9: 642.6, ICD 10: O15)

- Occurrence of severely high blood pressure (>170/110 mmHg) combined with either any preeclampsia-related ICD code (ICD 9: 642.4, 642.5; ICD 10: O11, O14) or abnormal liver function tests (LFTs) (AST > 64, ALT > 70).

This approach accurately identified the diagnosis in 88.5% of the confirmed preeclampsia cases. Additionally, the algorithm correctly identified 92.6% of cases with ambiguous documentation.

## Feature engineering

For time series vital sign measurements, we extracted key statistics by utilizing the tsfresh (version v0.20.2) library's extract_features function with EfficientFCParameters. Subsequently, we applied the calculate_relevance_table function to the training dataset to determine which features to include in the final dataset. Features that met significance (p values <0.05) included mean, maximum, minimum, median, number of readings, and the c3 feature with a lag of 2. The c3 statistic is a third-order correlation measure that assesses how three points in a time series, spaced at a fixed interval (or "lag"), are related to each other in a way that goes beyond simple linear correlations. We computed the c3 statistic with a lag of 2 to measure non-linearity by examining interactions between values separated by two steps. This choice balances capturing short-term fluctuations with accommodating the wide variation in readings between patients and cohorts, ensuring consistency and relevance across datasets where longer lags may have less meaningful interactions.

Furthermore, we implemented multiple techniques aimed at capturing blood pressure trends over time. We applied polynomial regression to each measurement to derive linear and quadratic coefficients, capturing temporal trends, and

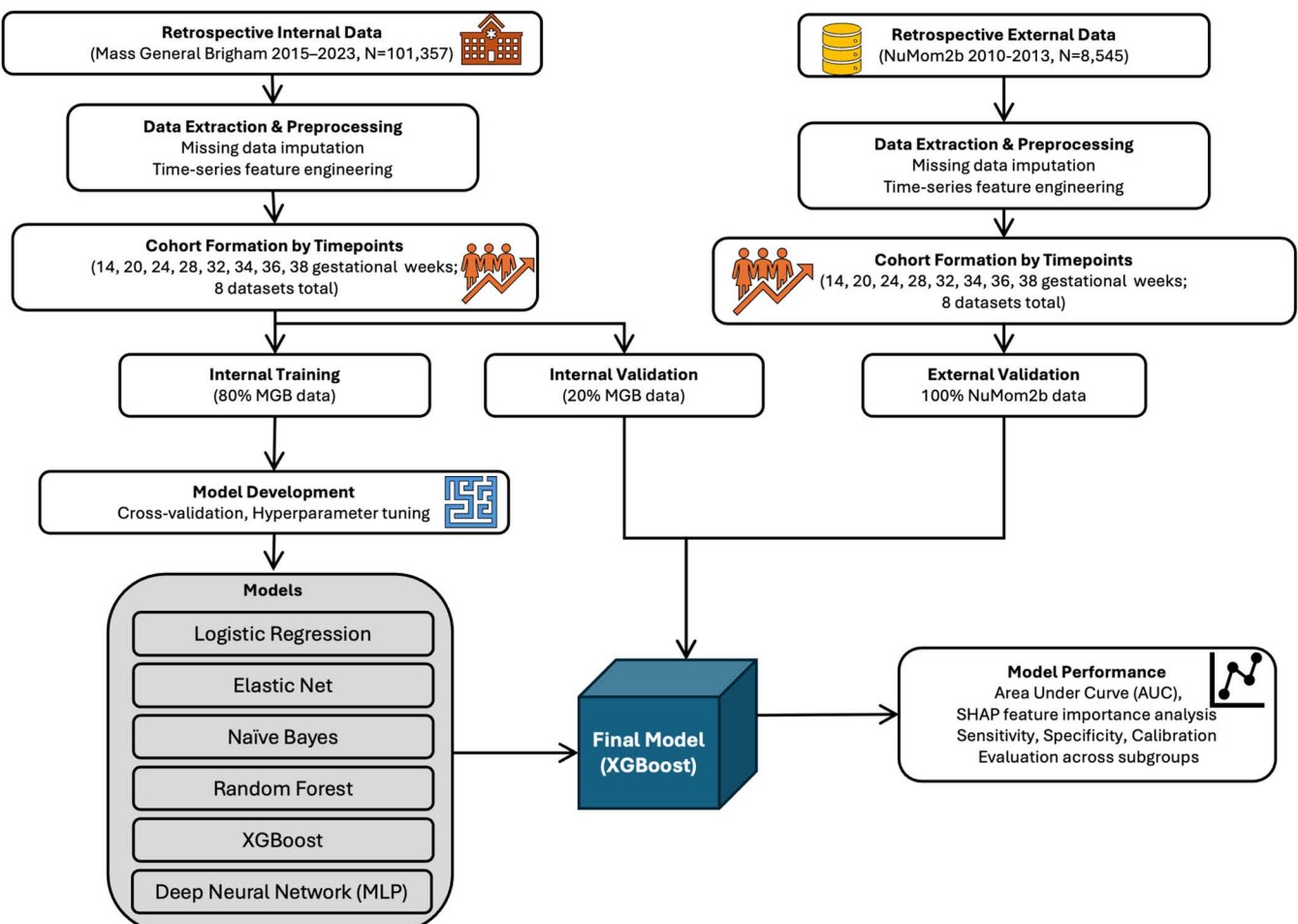

**Fig 1. Methodology flowchart.** The figure illustrates the sequential steps involved in developing and validating interpretable machine learning models for the longitudinal prediction of preeclampsia risk, highlighting data sources, preprocessing techniques, model building, and validation approaches.

integrated them as features in our analysis. We also discretized the time series into predefined bins based on the dataset cutoffs. For each bin, we used the minimum, maximum, and mean values and compared them across bins. Importantly, the polynomial regression features were designed to mitigate issues of collinearity inherent in simpler comparisons, which prevent their inclusion in linear models. The final features used in the models are listed in S1 Table.

## Model development

The study utilized EHR data from three Mass General Brigham hospitals for model development (80%) and a hold-out set (20%) for internal validation. The complete list of features used is shown in S1 Table. To prevent overlap, pregnancies were partitioned based on patient identifiers, ensuring that data from the same individual did not appear in both sets, which could artificially enhance performance metrics. External validation was conducted using the nuMoM2b dataset, an independent cohort not involved in model development. Hyperparameter optimization was performed through a 5-fold cross-validation within the development set (S2 and S3 Tables), and the model was then retrained on the full development dataset using the optimal hyperparameters. This final model was subsequently used to evaluate performance metrics on both the internal testing and external validation datasets.

We developed the following models, which have been used in prior research [11–13,18,26–28]: logistic regression, elastic net, naïve-Bayes, random forest, xgboost, and deep neural network. For deep neural networks, we employed a Multi-Layer Perceptron (MLP) with dynamically determined hidden layers and nodes per layer. For the logistic regression and elastic net models, we used all variables that were not colinear (Variance Inflation Factor < 10).

Hyperparameters, including the number of layers and nodes per layer, activation functions, optimization solvers, regularization strength, and learning rate strategy, were systematically tuned using Optuna (version 2.1.0). Using Optuna's TPE sampler [29], we ran ten iterations over the hyperparameter space to discover the best-fitting combination based on the mean of the AUC scores across all cross-validation folds (the hyperparameters are listed in S2 and S3 Tables). These parameters were then used to train the model on the entire training dataset and tested on the testing and external validation sets.

As the dataset was highly imbalanced, we investigated the following resampling methods: RandomOverSampler, which increases the minority class by randomly duplicating samples; RandomUnderSampler, which reduces the majority class by randomly removing samples; SMOTE (Synthetic Minority Over-sampling Technique), which generates synthetic samples for the minority class by interpolating between existing samples; and BorderlineSMOTE, a variation of SMOTE that focuses on generating synthetic samples near the decision boundary [30]. We determined that random oversampling provided the best performance as measured by the Area Under the Receiver Operating Curve (AUC), and that approach was used for the subsequent experiments.

We employed a systematic approach to data processing and algorithm development to identify and mitigate bias [31]. To evaluate potential racial and insurance bias, we used the Equality of Opportunity Difference (EOD) measure [32] to compare the true positive rates of our model across different sociodemographic groups. We determined the significance of these differences using bootstrapping with 1,000 iterations to calculate p-values.

We employed Shapley values [33], a game theoretic approach, to explain the output of any machine learning model. It connects optimal credit allocation with local explanations using the classic Shapley values from game theory and their related extensions. This method allows for both local (for individual patients) and global (for the model) interpretations. We used the SHAP Python package 0.41.0.

## Statistical analyses and definitions

For the descriptive analyses, we used all available EHR data from before conception to up to 6 weeks postpartum. Variables were expressed as median with interquartile range (IQR). Significance was determined using the Student's t-test for continuous variables and the Chi-squared test for categorical variables.

## Results

### Population characteristics and outcomes

We identified 104,893 deliveries at Mass General Brigham between May 2015 and May 2023. Of those, 101,357 met the additional selection criteria of having at least one visit with recorded information before 14 weeks of pregnancy and were subsequently used in our analysis (Fig 2). All three hospitals provided data for this study.

The overall incidence of preeclampsia was 6.1%, N = 6,160 (Table 3). Compared to normotensive patients, those with a diagnosis of preeclampsia included a significantly higher proportion of individuals with self-reported Black race (18.3% vs. 9.6%) and self-reported Hispanic ethnicity (18.5% vs 14.5%), respectively. In addition, patients with preeclampsia were more likely to have a family history of hypertension, higher maximal systolic and diastolic blood pressures, and higher weight gain during pregnancy compared with those who did not develop preeclampsia, P < 0.001. The remaining characteristics of both groups are summarized in Table 3.

The external nuMoM2b cohort included 8,545 patients, of which 6.3%, N = 536, developed preeclampsia (S4 Table). Similarly, those with preeclampsia, compared to those without, were more likely to be of Black race and had higher blood pressure and weight gain during pregnancy.

### Development of multiple predictive models during pregnancy

To develop predictive models of the preeclampsia risk over time, eight datasets were created at specified time points in pregnancy based on scheduled routine visits; the patient data from the nuMoM2b cohort were reserved for external validation. We developed six types of models: logistic regression, elastic net, naïve-Bayes, random forest, xgboost, and deep neural network at each of the eight different time points: 14, 20, 24, 28, 32, 34, 36, and 38 gestational weeks (Table 4). The earliest timepoint for which we developed each of the models was 14 weeks, as most patients had their first prenatal visit at that time, and commonly, new vital signs, detailed history, and baseline laboratory tests were obtained. We used the area under the curve (AUC), a widely used metric in machine learning that balances sensitivity and specificity, as the primary metric to evaluate the predictive power of the models.

We analyzed the AUC of each model using the hold-out MGB dataset. Overall, the models had AUC 0.71–0.77 (95%CI 0.69–0.79) at 14 gestational weeks. With the advancement of the gestational age, more data are collected from every

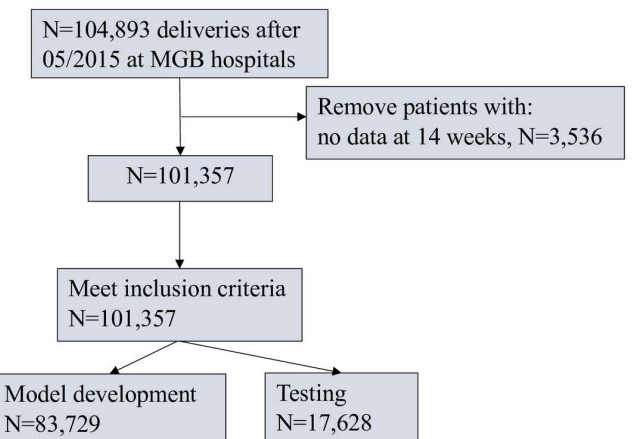
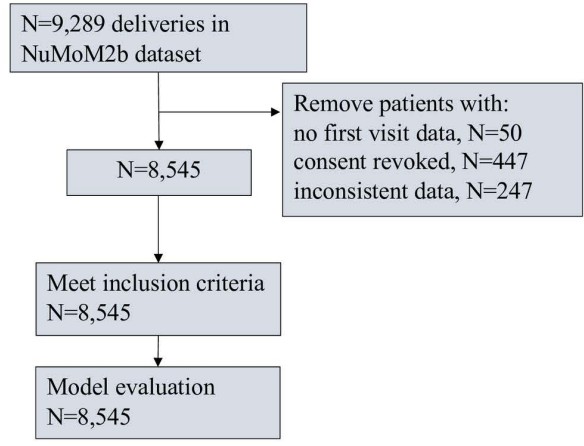

**Fig 2. Patient flow diagram.**

patient visit, and the performance of each model type increased; the highest AUC was at 28–36 weeks, with a range of 0.73–0.80 (95%CI 0.71–0.81). We externally validated the performance using data from nuMoM2b, which showed an AUC range of 0.57–0.70 (95%CI 0.55–0.73) (Table 5). This performance is lower compared to the held-out MGB dataset, yet it exhibits similar trends.

Compared with a logistic regression model using the risk stratification criteria based on the most current clinical guidelines, our models achieved better performance in both the hold-out MGB and nuMoM2b datasets (Fig 3). Interestingly, 25.5–47.1% of the false positive cases developed gestational hypertension, another hypertensive disorder of pregnancy. The majority of the false negative cases did not have any known risk factors.

**Table 3. Description of the Mass General Brigham cohort.**

| Characteristic | No preeclampsia (n = 95,197) | Preeclampsia (n = 6,160) | Total (n = 101,357) | P-value |
|---|---|---|---|---|
| Maternal age at delivery, y | 33.24 ± 4.79 | 33.34 ± 5.58 | 33.24 ± 4.84 | 0.10 |
| Self-reported race, White | 63163 (66.3%) | 3590 (58.3%) | 66753 (65.9%) | <0.001 |
| Self-reported race, Black | 9104 (9.6%) | 1117 (18.3%) | 10221 (10.2%) | <0.001 |
| Self-reported race, other | 12412 (13.0%) | 1023 (16.6%) | 13435 (13.3%) | <0.001 |
| Self-reported ethnicity, Hispanic | 13722 (14.5%) | 1127 (18.5%) | 14849 (14.8%) | <0.001 |
| Self-reported ethnicity, non-Hispanic | 81475 (85.6%) | 5033 (81.7%) | 86508 (85.3%) | <0.001 |
| Tertiary hospital | 70583 (74.1%) | 4951 (80.4%) | 75534 (74.5%) | <0.001 |
| Community hospital | 24614 (25.9%) | 1209 (19.6%) | 25823 (25.5%) | <0.001 |
| Gravidity | 1.95 ± 1.62 | 1.88 ± 1.81 | 1.94 ± 1.64 | 0.003 |
| Parity | 0.63 ± 0.91 | 0.51 ± 0.96 | 0.62 ± 0.91 | <0.001 |
| Nulliparous | 54200 (56.9%) | 4233 (68.7%) | 58433 (57.7%) | <0.001 |
| In vitro fertilization | 3939 (4.1%) | 393 (6.4%) | 4332 (4.3%) | <0.001 |
| Multiple gestation | 2812 (3.0%) | 487 (7.9%) | 3299 (3.3%) | <0.001 |
| Gestational age at delivery, weeks | 273.33 ± 14.21 | 259.87 ± 20.71 | 272.52 ± 15.04 | <0.001 |
| BMI at delivery, kg/m² | 30.81 ± 5.71 | 34.58 ± 7.39 | 31.06 ± 5.90 | <0.001 |
| Weight at delivery, kg | 82.29 ± 16.01 | 91.74 ± 20.86 | 82.88 ± 16.52 | <0.001 |
| Maximal SBP during pregnancy, mmHg | 141.86 ± 15.22 | 175.26 ± 16.93 | 143.90 ± 17.28 | <0.001 |
| Maximal DBP during pregnancy, mmHg | 87.29 ± 10.41 | 104.88 ± 12.06 | 88.37 ± 11.33 | <0.001 |
| Family history of chronic hypertension | 35918 (37.7%) | 2868 (46.6%) | 38786 (38.3%) | <0.001 |
| Family history of preeclampsia | 2069 (2.2%) | 151 (2.5%) | 2220 (2.2%) | 0.16 |
| Past history of chronic hypertension | 5357 (5.6%) | 1835 (29.8%) | 7192 (7.1%) | <0.001 |
| Past history of gestational hypertension | 3331 (3.5%) | 498 (8.1%) | 3829 (3.8%) | <0.001 |
| Past history of preeclampsia | 2127 (2.2%) | 531 (8.6%) | 2658 (2.6%) | <0.001 |
| Past history of preterm delivery (<37 weeks) | 4063 (4.3%) | 1440 (23.4%) | 5503 (5.4%) | <0.001 |
| Antihypertensive medications throughout pregnancy and 6 weeks postpartum | 40288 (42.3%) | 5063 (82.2%) | 45351 (44.7%) | <0.001 |
| Headaches during pregnancy and 6 weeks postpartum | 4075 (4.3%) | 700 (11.4%) | 4775 (4.7%) | <0.001 |
| Gestational diabetes | 13402 (14.1%) | 1260 (20.5%) | 14662 (14.5%) | <0.001 |
| Proteinuria | 578 (0.6%) | 3747 (60.8%) | 4325 (4.3%) | <0.001 |
| Maximal uric acid during pregnancy | 4.76 ± 1.35 | 5.82 ± 1.52 | 5.00 ± 1.46 | <0.001 |
| SGA or IUGR | 8099 (8.5%) | 891 (14.5%) | 8990 (8.9%) | <0.001 |

Mean ± standard deviation for continuous variables; n (%) for categorical variables; p-values for continuous variables were calculated using the Student's t-test; for categorical variables based on the Chi-squared test.

Abbreviations: SBP, systolic blood pressure; DBP, diastolic blood pressure, BMI, body mass index, SGA, small for gestational age, IUGR, intrauterine growth restriction.

## Bias evaluation in different subgroups

We also assessed potential racial and insurance bias in our model by comparing performance across different sociodemographic groups, including White, Black, Hispanic, and Asian populations, as well as different insurance groups (private and public). We found no significant differences in model performance among these groups (S5 Table).

## Model calibration

To better understand the models' stability and performance, we investigated their behavior across different probability cutoffs for classifying patients as positive or negative based on their risk for developing preeclampsia (S6 Table). The models maintained high specificity across all cutoff values. When the cutoff value was decreased from 0.5 to 0.4, there was a 10–15% increase in sensitivity, indicating that more true positive cases were correctly identified. However, this increase in sensitivity came with a comparable decrease in specificity, identifying more false positives.

## Comparison to the current standard of care

We further explored the performance of the xgboost models in comparison with the current standard of care based on the presence of clinical risk factors.

Early in pregnancy, at 14 weeks, when aspirin prophylaxis is indicated and may favorably alter the course of the disease, the novel xgboost model identified 5,462 (31.0%) of all patients at risk, while 3,618 (20.5%) of all patients were identified using the ACOG risk stratification criteria using the MGB held-out test dataset (Fig 4). Of those, our xgboost model correctly identified 732 (67.8%) of the patients who subsequently developed preeclampsia; in contrast, the traditional method identified 580 (53.7%) patients.

For the timepoint closest to delivery, the xgboost correctly identified 331 (67%) of the cases, while the risk stratification logistic model identified 400 (81.0%). However, the latter had 8677 (58.4%) at-risk individuals, while our xgboost had only 3487 (23.5%). The classification performance of both types of models across different gestational ages is shown in Fig 5.

## Time course and relationships of the most predictive variables

Using Shapley additive explanations method, we evaluated the most predictive variables in the xgboost model, over the eight time points (Fig 6). In early pregnancy, at 14 weeks, the most predictive features were chronic hypertension and

Table 4. Internal validation area under the curve results for models at different gestational ages.

| Gestational Weeks | Logistic regression | Elastic net | Naïve bayes | Random Forest | Xgboost | Neural Network |
|---|---|---|---|---|---|---|
| 14 | 0.75 (0.73 - 0.76) | 0.74 (0.72 - 0.75) | 0.71 (0.69 - 0.73) | 0.76 (0.75 - 0.78) | 0.76 (0.75 - 0.78) | 0.77 (0.76 - 0.79) |
| 20 | 0.74 (0.72 - 0.75) | 0.74 (0.72 - 0.75) | 0.72 (0.71 - 0.74) | 0.77 (0.75 - 0.78) | 0.78 (0.77 - 0.80) | 0.78 (0.77 - 0.80) |
| 24 | 0.73 (0.72 - 0.75) | 0.73 (0.72 - 0.74) | 0.73 (0.72 - 0.75) | 0.78 (0.76 - 0.79) | 0.78 (0.77 - 0.79) | 0.79 (0.78 - 0.80) |
| 28 | 0.74 (0.72 - 0.76) | 0.73 (0.71 - 0.75) | 0.75 (0.73 - 0.76) | 0.79 (0.77 - 0.80) | 0.80 (0.78 - 0.81) | 0.80 (0.78 - 0.81) |
| 32 | 0.74 (0.72 - 0.76) | 0.73 (0.71 - 0.75) | 0.75 (0.73 - 0.76) | 0.78 (0.77 - 0.80) | 0.80 (0.78 - 0.82) | 0.80 (0.79 - 0.81) |
| 34 | 0.74 (0.72 - 0.75) | 0.73 (0.72 - 0.75) | 0.74 (0.72 - 0.76) | 0.78 (0.76 - 0.80) | 0.80 (0.79 - 0.81) | 0.80 (0.78 - 0.81) |
| 36 | 0.72 (0.71 - 0.74) | 0.72 (0.70 - 0.73) | 0.73 (0.71 - 0.75) | 0.78 (0.77 - 0.80) | 0.79 (0.78 - 0.81) | 0.80 (0.78 - 0.82) |
| 38 | 0.71 (0.69 - 0.73) | 0.71 (0.69 - 0.73) | 0.75 (0.73 - 0.77) | 0.77 (0.75 - 0.80) | 0.79 (0.77 - 0.81) | 0.79 (0.78 - 0.81) |

kidney disease. With advancing gestational age, the highest contributing features become systolic and diastolic blood pressure. Similarly, in early pregnancy, the category that contributed the most to the model predictions was medical history; with advancing gestational age, the vital signs and laboratory results became more contributory (S1 Fig).

In addition to exploring the contribution of individual features to the model's prediction, Shapley values can highlight the interactions between different variables (S2 Fig). The relationships between some variables follow non-linear patterns; for example, age is a risk factor if the patient is < 20 years or >35 years old. Age also interacts with other features, such as nulliparity and White race, in a nonlinear fashion.

## Discussion

We developed models predicting preeclampsia risk using machine learning methods at eight different timepoints during pregnancy. With the advancement of pregnancy and the increase in the gestational weeks, the models get progressively more accurate, from an area under the curve (AUC) of 0.76 to 0.80; the highest predictive value, AUC 0.80, was achieved at 28–36 weeks. The most predictive features were systolic and diastolic blood pressure and interpregnancy interval. As

**Table 5. External validation area under the curve results for models at different gestational ages.**

| Gestational Weeks | Logistic regression | Elastic net | Naïve bayes | Random Forest | Xgboost | Neural Network |
|---|---|---|---|---|---|---|
| 14 | 0.58 (0.55 - 0.60) | 0.58 (0.56 - 0.60) | 0.64 (0.61 - 0.66) | 0.66 (0.64 - 0.68) | 0.64 (0.61 - 0.66) | 0.66 (0.63 - 0.67) |
| 20 | 0.58 (0.56 - 0.60) | 0.58 (0.55 - 0.60) | 0.65 (0.62 - 0.68) | 0.66 (0.64 - 0.69) | 0.66 (0.64 - 0.69) | 0.66 (0.64 - 0.69) |
| 24 | 0.57 (0.55 - 0.60) | 0.57 (0.54 - 0.60) | 0.66 (0.64 - 0.69) | 0.68 (0.65 - 0.70) | 0.67 (0.65 - 0.69) | 0.68 (0.65 - 0.70) |
| 28 | 0.57 (0.55 - 0.60) | 0.57 (0.55 - 0.60) | 0.67 (0.65 - 0.70) | 0.69 (0.67 - 0.71) | 0.70 (0.67 - 0.72) | 0.69 (0.67 - 0.72) |
| 32 | 0.58 (0.56 - 0.60) | 0.58 (0.55 - 0.60) | 0.68 (0.65 - 0.70) | 0.69 (0.67 - 0.72) | 0.70 (0.67 - 0.72) | 0.70 (0.68 - 0.73) |
| 34 | 0.58 (0.55 - 0.61) | 0.58 (0.55 - 0.61) | 0.68 (0.65 - 0.70) | 0.69 (0.67 - 0.71) | 0.70 (0.68 - 0.73) | 0.70 (0.68 - 0.72) |
| 36 | 0.58 (0.55 - 0.62) | 0.58 (0.55 - 0.61) | 0.69 (0.66 - 0.71) | 0.70 (0.67 - 0.73) | 0.70 (0.67 - 0.72) | 0.70 (0.67 - 0.73) |
| 38 | 0.60 (0.57 - 0.64) | 0.60 (0.57 - 0.63) | 0.68 (0.65 - 0.73) | 0.68 (0.65 - 0.72) | 0.69 (0.66 - 0.73) | 0.67 (0.63 - 0.71) |

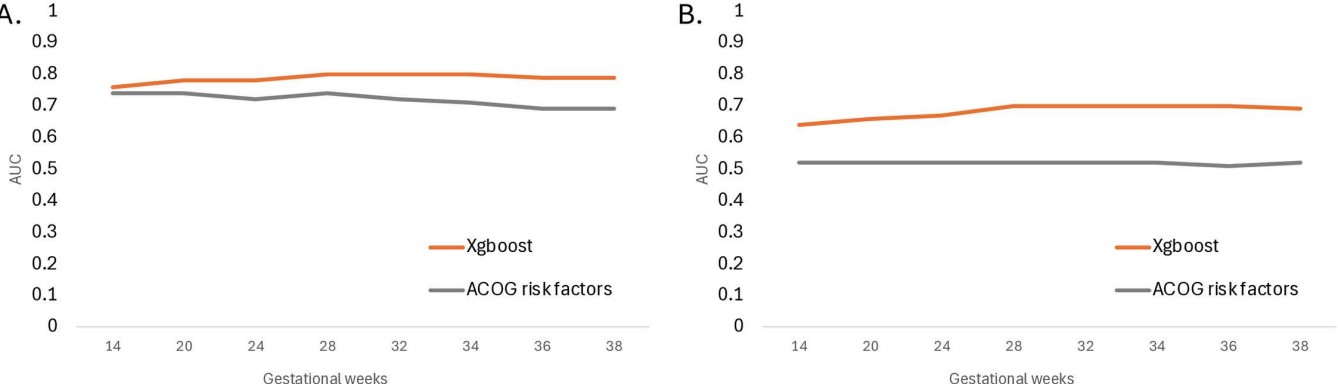

**Fig 3. Comparison of the xgboost and logistic regression model with clinical risk criteria (ACOG).** A. Performance in the held-out Mass General Brigham dataset. B. Performance in the Nulliparous Pregnancy Outcomes Study: Monitoring Mothers-to-Be (nuMoM2b).

gestational age advanced, the relative significance of systolic and diastolic blood pressure compared to other predictors increased significantly. Compared with the current standard of care, our models have higher predictive power and would allow earlier and more precise identification of patients at risk for preeclampsia. However, the benefits of the novel models need to be balanced with the need for surveillance in a larger at-risk population.

The importance of screening for preeclampsia throughout pregnancy has been emphasized [1], and to date, multiple predictive models have been developed [11–13,18,26–28]. Most studies include clinical data as well as biomarkers, such as serum placental growth factor [10,28], pregnancy-associated plasma protein A [10,28], and uterine artery pulsatility index [8,10,28]. However, these tests are not routinely performed in all clinical settings, and thus, these models have limited application. We aimed to develop an accurate tool that can be applied throughout pregnancy using data routinely collected during prenatal visits. In this way, our approach can be implemented in any EHR.

The current study is the largest investigation to date based on the cohort size, time points, depth of predictor variables, and types of machine learning models. We achieved high predictive power using our models, which are comparable to or better than other studies using EHR [8,12,13,18]. These prior studies developed models from all available EHR data, which may lead to overfitting [12,18], did not investigate multiple machine learning models [12,13], and the datasets have high missingness [13]. We overcome the limitations of prior work by including only those variables that are clinically relevant for the diagnosis of preeclampsia in our models; we anticipate that this approach will improve cross-institutional reproducibility.

In light of AI bias and the selective disadvantage of individuals with self-reported Black race, multiple strategies have been developed to ameliorate the perpetuation of racial bias [26,28]. Similar to other studies [9,11], we also find that Black individuals are at higher risk for preeclampsia. We investigated the role of race, ethnicity, and insurance status in combination with other sociodemographic variables and employed best practices to develop highly performant and equitable tools. We achieved equitable performance with our models in multiple sociodemographic groups; however, we cannot exclude additional sources of bias. Elevated preeclampsia risk remains in Black individuals with higher income and socioeconomic status, suggesting the role of stress, innate biologic factors (e.g., genetic predisposition), and additional factors associated with systemic racism [34]. Further work in elucidating the etiologies of disparities will be important for the successful and equitable implementation of our models.

External validation is a major concern in modeling studies [27], and most published models have lower predictive power in external validation due to changes in the feature definitions, missing data, and overfitting of the model in the original

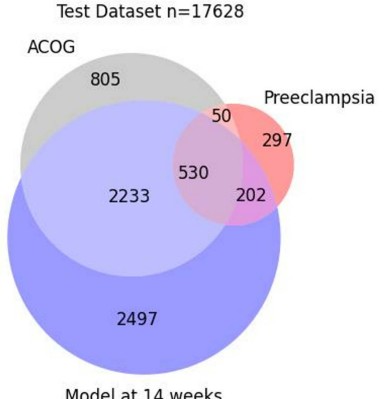

**Fig 4. Comparison between the xgboost model and the current standard of care as determined by ACOG guidelines.** Venn diagram of all individuals at risk for preeclampsia in the test dataset (N = 17,628); of those, individuals at risk predicted using the ACOG criteria (ACOG) are in grey, the individuals at risk predicted using the xgboost model (model) at 14 weeks are in purple, and those who developed preeclampsia are in pink.

population [27]. For external validation, we selected nuMoM2b, a large, prospectively collected dataset from multiple sites in the US. In that dataset, we report similar trends but lower performance compared to the model development dataset, which is likely due to differences in the study populations, changes in the criteria for diagnosis of preeclampsia and clinical practice after the nuMoM2b data were collected [1], and limited data collected only at three time points. Compared to the external dataset, our dataset also includes individuals with multiple gestations and prior pregnancies, and data collected at least eight visits during pregnancy. This enabled more comprehensive and granular feature selection at the model development stage.

Given the robustness of our large cohort, we explored numerous features and interactions. While risk factors have been well investigated [35], only a few prior studies have analyzed all patient EHR data during pregnancy in its entirety [11,12]. We demonstrated the strong effect on preeclampsia risk of known predictors, such as a past medical history of preeclampsia, diabetes, and chronic kidney disease [9,13,26]. The incorporation of time-series variables improves the accuracy of predictions [11,12], and we replicate prior findings of heightened preeclampsia risk with excessive weight gain [11,35,36] and steep patterns of blood pressure increase [11,12,37] during pregnancy. Due to the lack of evidence for the relative role of individual risk factors and the interaction between those risk factors [35], our study adds substantial results from clinical practice to inform future predictive tools and guidelines. The risk of preeclampsia is higher in the extremes of age [38], and also in pregnancies with short and long interpregnancy intervals [39]. The connection with red blood cell

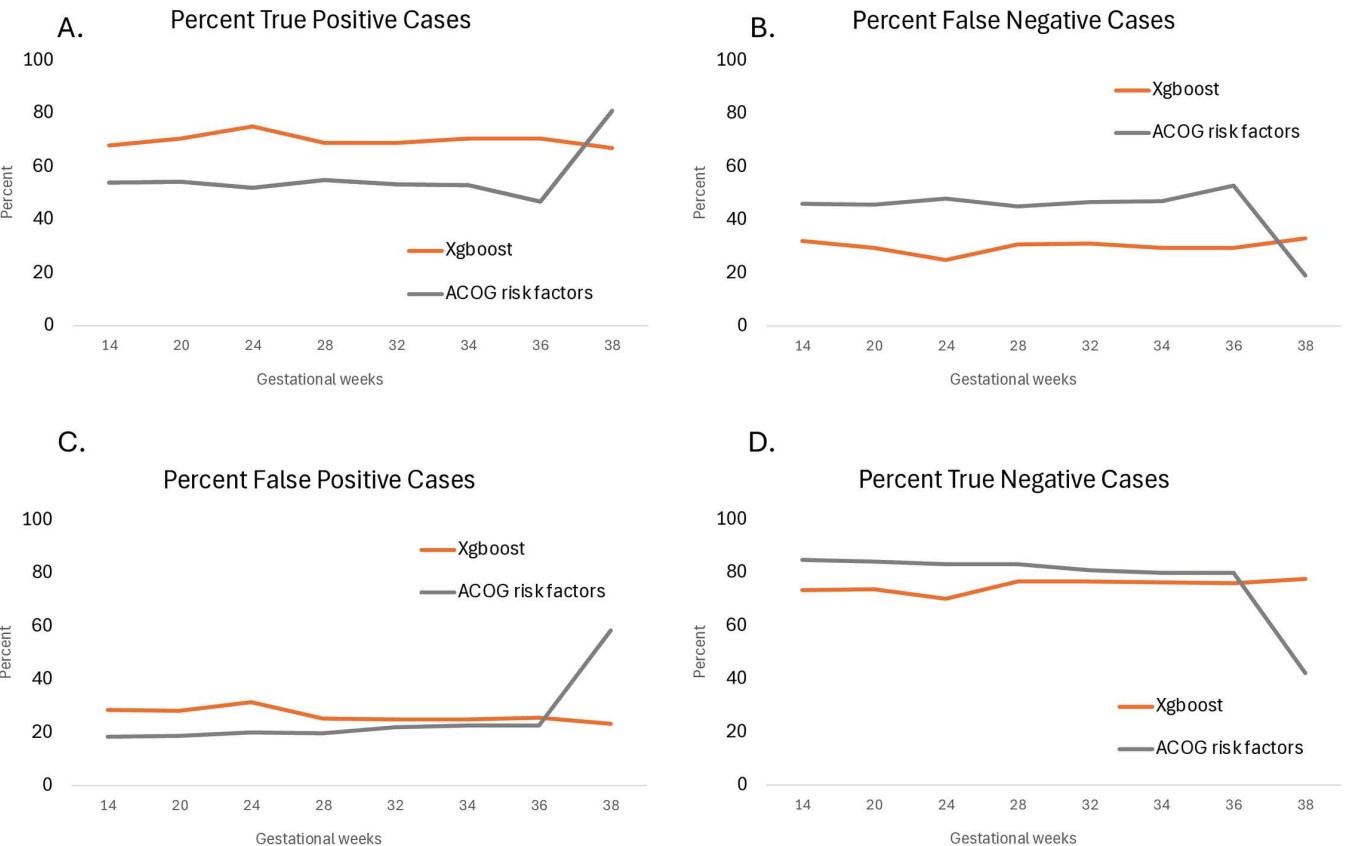

**Fig 5. Classification of model performance across gestational age.** A. Percent true positive cases out of those who developed preeclampsia. B. Percent false negative cases out of those who developed preeclampsia. C. Percent at-risk individuals out of those who did not develop preeclampsia. D. Percent low-risk individuals out of those who did not develop preeclampsia.

count is novel and may be related to maternal hemoconcentration [40]. As preeclampsia is a heterogeneous disorder, further exploring the interaction between variables has the potential to identify sub-phenotypes of preeclampsia that may benefit from a more personalized approach to prophylaxis and treatment [41].

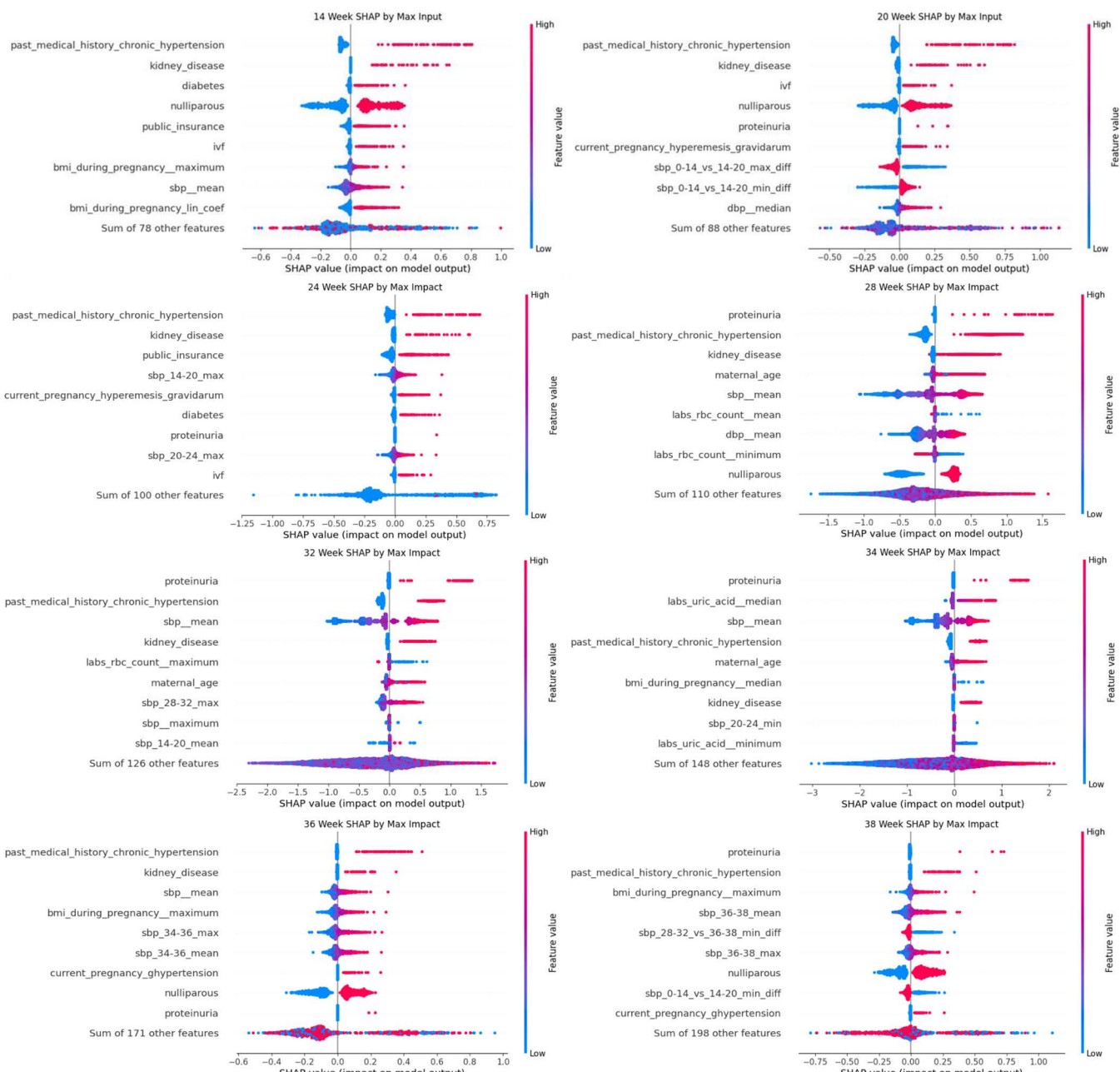

**Fig 6. Shapley plots for the best-performing preeclampsia prediction model at every timepoint.** Shapley values plot showing the impact of individual feature values on the model output. Red points represent high feature values, while blue points indicate low feature values. Points on the right side of the X-axis contribute to a higher likelihood of the output class being present.

Our models identify more at-risk patients compared to the standard of care. As a result, the implementation of our models would result in a higher number of patients eligible for prophylaxis and surveillance because there is no definitive therapy for preeclampsia other than delivery. The current guidelines recommend starting aspirin prophylaxis early in pregnancy in patients for high-risk individuals [1] which can decrease the incidence and severity of early-onset preeclampsia [42,43]. Low-dose aspirin is considered safe in pregnancy, and the studies of aspirin-related bleeding risk are equivocal [44]. Nevertheless, the adverse effects of aspirin prophylaxis in a high number of at-risk patients need to be carefully balanced with the risk of developing early-onset preeclampsia, the associated maternal morbidity, and iatrogenic preterm birth.

Our predictive modeling approach offers significant promise for integration into public health systems globally, potentially improving maternal and neonatal health outcomes through earlier identification and proactive management of preeclampsia. By relying solely on routinely collected clinical data available through standard EHR data, this model would be feasible to implement across diverse healthcare settings, including low-resource environments. However, several challenges must be addressed to facilitate broad global adoption. Limited infrastructure and inconsistent EHR availability in low- and middle-income countries could present significant barriers. To mitigate these challenges, implementation strategies could include developing simplified mobile-health (mHealth) adaptations, leveraging widely available technologies such as smartphones, and creating streamlined, low-cost data capture systems. Additionally, ensuring model transferability and robustness across diverse populations requires local validation and potential retraining of predictive models to account for population-specific risk factors and variations in clinical practice.

Prospective validation represents a critical next step to ensure the clinical applicability and utility of our predictive models. Future research should involve deploying this model prospectively in clinical practice within diverse healthcare settings, carefully assessing real-world predictive performance, clinician usability, patient outcomes, and cost-effectiveness. Ultimately, such validation is essential to establish confidence among clinicians and policymakers, driving widespread adoption and ensuring tangible improvements in maternal and neonatal health outcomes.

## Limitations

Our study has several limitations. In some cases, the diagnosis of preeclampsia may be challenging to establish as there may be no clear tests that distinguish between gestational hypertension, chronic hypertension, and superimposed preeclampsia. There is also a degree of uncertainty in the diagnostic criteria definitions [1,45]. Traditionally, most retrospective and modeling studies have used disease codes [11], which may be inaccurate [46]. To improve the accuracy of the outcome, we developed and manually validated an algorithm based on structured data. In addition, as some patients utilized home blood pressure measurements, those values were not available in the EHR and, thus, were not included in the models. In the future, integrating home blood pressure monitoring may improve the predictive power of models. We also acknowledge that the American College of Obstetricians and Gynecologists (ACOG) risk factors [1] may not be an ideal comparative model to our models; however, this approach has been used by others [12], and most modeling studies have insufficient data to reproduce the results [27]. Lastly, we used retrospective data to test and externally validate our models, and further prospective validation is needed to evaluate the true predictive power.

## Conclusion

By using routinely collected data during scheduled prenatal visits, we demonstrate that accurate prediction of preeclampsia can be achieved. This design would allow risk assessment utilizing these models at every visit throughout pregnancy with updates in risk prediction, resulting in a more accurate ascertainment of individuals at risk who would benefit from prophylactic measures or increased surveillance. In the future, as better testing or prophylaxis become available, these AI tools can be used to select the group of individuals who would benefit the most.

## Supporting information

**S1 File.** Fig S1. Change in percentage of feature group contribution for model prediction throughout pregnancy. Fig S2. Interaction between individual features using Shapley values. Comparison between the xgboost model and the current standard of care as determined by ACOG guidelines. Table S1. Features used in the models. Table S2. Hyperparameter space for each model. Table S3. Best performing parameters for each model. Table S4. Characteristics of the external NuMoM2b dataset. Table S5. Equality of Opportunity results. Table S6. Model calibration.
(DOCX)

## Author contributions

**Conceptualization:** Raphael Y Cohen, David W Bates, Kathryn J Gray, Vesela Kovacheva.

**Data curation:** Raphael Y Cohen, Nolan Wheeler, Ricardo Kleinlein, John Rigoni.

**Formal analysis:** Braden W Eberhard, Ricardo Kleinlein.

**Funding acquisition:** Vesela Kovacheva.

**Investigation:** Braden W Eberhard, Raphael Y Cohen, David W Bates, Kathryn J Gray.

**Methodology:** Braden W Eberhard, Nolan Wheeler, Ricardo Kleinlein, John Rigoni, Kathryn J Gray, Vesela Kovacheva.

**Project administration:** Vesela Kovacheva.

**Software:** Braden W Eberhard, Raphael Y Cohen, Nolan Wheeler, Ricardo Kleinlein, John Rigoni.

**Supervision:** Kathryn J Gray, Vesela Kovacheva.

**Validation:** Braden W Eberhard.

**Writing – original draft:** Braden W Eberhard, Vesela Kovacheva.

**Writing – review & editing:** Braden W Eberhard, Raphael Y Cohen, Nolan Wheeler, Ricardo Kleinlein, John Rigoni, David W Bates, Kathryn J Gray, Vesela Kovacheva.

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
