## [Decision Letter · Decision Letter 0]

5 Mar 2025

Dear Dr. Kovacheva,

Thank you for submitting your manuscript to PLOS ONE. After careful consideration, we feel that it has merit but does not fully meet PLOS ONE’s publication criteria as it currently stands. Therefore, we invite you to submit a revised version of the manuscript that addresses the points raised during the review process.

**Epidemiological Context in the Introduction**

Include global data on the incidence and prevalence of preeclampsia, as well as additional potential factors influencing its development. This will help reinforce the study’s relevance.**Data Availability and Methodology Visualization**Clarify whether the data used in this study will be made available to ensure transparency and reproducibility. If possible, we suggest depositing the data in a public repository such as Zenodo, Figshare...Include a flowchart of the methodology to visually illustrate the study’s steps, improving readability.**Dedicated Section for Related Works**Create a specific section to discuss similar studies, detailing the key differences between the proposed model and previous approaches.Consider the need to include references to relevant studies, such as **"ILITIA: telehealth architecture for high-risk gestation classification"** , highlighting differences between machine learning-based models and specialized architectures that do not use this approach.**Discussion on Public Health Applicability**Expand the discussion on how this technology can be integrated into public health systems globally, particularly to ensure greater equity in early diagnosis and treatment.Address potential challenges in adopting this tool in countries with varying healthcare access and discuss possible strategies for implementation.**Prospective Validation**Include a brief discussion on future strategies for prospective validation of the model and its importance in consolidating clinical applicability.**Improving Accessibility and Clarity**Review the text to enhance its accessibility for a broader audience, adding further explanations where needed and, if applicable, providing supplementary material for technical concepts.

We look forward to receiving your revised manuscript.

Kind regards,

Zenewton André da Silva Gama, Ph.D.

Academic Editor

PLOS ONE

Additional Editor Comments:

These improvements will enhance the clarity and impact of the article. We look forward to receiving the revised version.

Reviewers' comments:

Reviewer's Responses to Questions

**Comments to the Author**

1. Is the manuscript technically sound, and do the data support the conclusions?

Reviewer #1: Yes

Reviewer #2: Yes

2. Has the statistical analysis been performed appropriately and rigorously?

Reviewer #1: Yes

Reviewer #2: Yes

3. Have the authors made all data underlying the findings in their manuscript fully available?

Reviewer #1: No

Reviewer #2: Yes

4. Is the manuscript presented in an intelligible fashion and written in standard English?

Reviewer #1: Yes

Reviewer #2: Yes

Reviewer #1: The manuscript “Development and validation of an interpretable longitudinal preeclampsia risk prediction using machine learning” presents a study that highlights the development of a machine learning-based tool to longitudinally predict the risk of preeclampsia. This is a relevant topic, mainly because it can contribute to minimizing risks during pregnancy and childbirth - a factor dominated by the quality of prenatal care.

The research developed by the authors presents a new approach to predicting preeclampsia that allows doctors to identify patients at risk early and provide personalized predictions throughout pregnancy - something desired, for example, for women with pregnancies considered high risk in primary health care.

Recommendations

[1] Introduction

The text presented in the introduction is quite clear and objective, and is very well written. However, it would be important for the authors to include more global epidemiological data on preeclampsia. What is the incidence and prevalence of this health problem in pregnant women worldwide? In addition to the factors presented by the authors, are there other factors that may be decisive for the development of preeclampsia?

I believe that this information will make the text even more qualified and will better justify the contributions of the research.

[2] Methodology

The methodology is very well documented, which guarantees the reproducibility of the steps used. However, I would like to know if the authors will make the data used in this research available? This is important to ensure greater transparency and reproducibility of the experiment. The data can be made available in a supplementary file or a public domain repository, for example Zenodo.

I believe that including a flowchart with the sequence of steps used in the methodology will further qualify the text and illustrate in a more visual way the entire path followed by the authors during the experiments.

[3] Related works

In the introduction, the authors present some related works and also manage to critically discuss some of these works. However, I consider it important to create a specific section for this. It would be interesting for the authors to discuss this topic in more depth, what are the similar works and what is the main difference between these works and also between what is being proposed by the authors? This is very important, as it better qualifies the research and highlights its contributions. The main contribution cannot only use machine learning; in my opinion, the main contribution is the predictive capacity and its application, especially in public health.

For example, the work “ILITIA: telehealth architecture for high-risk gestation classification” works on the classification of high-risk pregnancies - pregnant women with high-risk pregnancies, but it does not use machine learning. It is a specialist architecture that applies a validated classification protocol that has great application in public health, in particular by carrying out this screening process remotely, through telediagnosis and teleconsultation.

[4] Discussions

The results presented in the manuscript are relevant. So I expected a discussion that addressed issues related to public health. Preeclampsia is a global health problem, so how can women from all over the world have access to this type of health care? How can this type of technology be included in public health so that more health professionals can have access to it and apply it to their patients?

These are more important challenges than those of technology, because the technological challenges are being solved, but expanding access and improving equity in the health of the population does not seem to be on the list of priorities of countries - as is the case in the United States, for example, which has the technology, but the health system is still very expensive and fragile - it is not inclusive.

Reviewer's consideration

I would like to congratulate the authors for their excellent work, and I am grateful for the opportunity to review this manuscript.

The work is of good quality and deserves to be published, but it needs minor adjustments first.

Reviewer #2: This manuscript describes the development of a longitudinal preeclampsia prediction tool, and the clinical impact of the usage of the tool in the context of current literature.

The manuscript has incorporated a robust and multi‐faceted methodology which makes it technically sound. The study has a comprehensive data source from three hospitals. The paper has been further strengthened with the external validation using the nuMoM2b dataset. The study demonstrates a thorough exploration of predictive modelling techniques. Shapley values have been used for interpretability. Clear Performance Metrics are evidenced by reporting AUCs with confidence intervals at eight gestational time points. The manuscript has not found any significant differences in performance based on race and ethnicity which is important for clinical applicability of the model.

The data largely support the conclusions from the study. The predictive model shows promise for early and personalized risk identification for preeclampsia to help in treatment of the patients.

The main claims of the paper is that it presents as a novel longitudinal prediction tool to predict the risk of preeclampsia at multiple gestational time points using a diverse collection of data. The tool demonstrated internally acceptable performance and has also been externally validated using the nuMoM2b cohort, supporting its potential generalizability.

The significance of the claims are critical as it addresses a critical gap in current clinical tools, which can miss up to 66% of patients at risk. The model has integrated a broad range of data types with advanced machine learning techniques which can provide data-driven, personalized medicine in obstetrics.

The claims in the paper are generally placed within the context of known limitations in the field, and the authors appear to have treated the literature fairly.

The analyses support the claims regarding improved preeclampsia risk prediction. But further evidence in the form of prospective studies are needed. This would provide a greater validation of the tool’s effectiveness and clinical impact.

This is a retrospective study and hence there is no prespecified trial protocol against which deviations can be assessed.

The details of the methodology has been incorporated in the paper - including study population, selection criteria, the time periods, eight gestational points for data collection, the types of data used, the modelling techniques used. This is sufficient to allow the experiments to be reproduced.

The limitations section is well-articulated with the authors rightly pointing out the diagnostic ambiguity due to the overlapping features with gestational and chronic hypertension. The exclusion of home blood pressure measurements as it is unavailable in the EHR is clearly stated. This suggests to integrate home monitoring in future work indicates a proactive approach to enhancing predictive power. The study is based on retrospective data and that prospective validation is needed as noted in the limitation section.

The manuscript is generally well organized and written in standard English following a clear structure by describing the background, methodology, results, and conclusions in a logical sequence. Including brief explanations or supplementary material could further enhance accessibility and understandability for a broader audience.

**Do you want your identity to be public for this peer review?** For information about this choice, including consent withdrawal, please see our Privacy Policy

Reviewer #1: **Yes: ** Ricardo Valentim

Reviewer #2: **Yes: ** Dr.Anuradha Pichumani

---

## [Author Response · Author response to Decision Letter 1]

21 Mar 2025

Dear Editor and Reviewers,

Thank you for providing thoughtful comments and suggestions to improve our manuscript, “Development and validation of an interpretable longitudinal preeclampsia risk prediction using machine learning.” We have carefully addressed each point raised by the Editor and Reviewers. Below, please see our point-by-point response. Where relevant, we have indicated the corresponding revisions in the manuscript (in italics).

Editor Comments:

Thank you for submitting your manuscript to PLOS ONE. After careful consideration, we feel that it has merit but does not fully meet PLOS ONE’s publication criteria as it currently stands. Therefore, we invite you to submit a revised version of the manuscript that addresses the points raised during the review process.

1. Epidemiological Context in the Introduction

o Include global data on the incidence and prevalence of preeclampsia, as well as additional potential factors influencing its development. This will help reinforce the study’s relevance.

Response 1.1: We appreciate the Editor and Reviewer #1’s suggestion. To address this point, we have revised the Introduction to include a more comprehensive epidemiological context. Specifically, we added global estimates on the incidence and prevalence of preeclampsia and expanded the discussion of additional factors influencing its development, such as genetic, environmental, and socioeconomic determinants. This addition reinforces the global relevance of our study and highlights the importance of our prediction models in a broader epidemiological context.

We have added the following text to the Introduction:

Worldwide, preeclampsia accounts for up to 10 million cases annually and is responsible for approximately 26% of maternal deaths in low- and middle-income countries[4, 5].

Additional potential factors influencing preeclampsia include genetic predisposition, maternal immunologic and inflammatory responses, environmental factors such as air pollution and dietary intake, and socioeconomic determinants including limited healthcare access and chronic stress [1, 4, 5].

2. Data Availability and Methodology Visualization

o Clarify whether the data used in this study will be made available to ensure transparency and reproducibility. If possible, we suggest depositing the data in a public repository such as Zenodo, Figshare...

Response 1.2: Thank you for highlighting the importance of data transparency. As indicated in our original submission, the data utilized in this study originate from the entire electronic health records of Mass General Brigham patients. Due to patient privacy, confidentiality concerns, and institutional data-sharing policies, we are unable to publicly deposit these patient-level data. However, the data used for external validation (nuMoM2b cohort) are publicly available through the NICHD DASH Data and Specimen Hub (https://dash.nichd.nih.gov/). For reproducibility purposes, we have provided a detailed methodology, including clearly specified algorithms, parameters, and hyperparameters (see supplementary information), enabling replication of our modeling approach on other datasets.

o Include a flowchart of the methodology to visually illustrate the study’s steps, improving readability.

We appreciate this suggestion and agree that a visual representation of our study workflow will enhance clarity. We have now included a new figure (Figure 1) depicting a detailed flowchart of our methodology, clearly illustrating the main steps from data extraction to model validation.

3. Dedicated Section for Related Works

o Create a specific section to discuss similar studies, detailing the key differences between the proposed model and previous approaches.

o Consider the need to include references to relevant studies, such as "ILITIA: telehealth architecture for high-risk gestation classification", highlighting differences between machine learning-based models and specialized architectures that do not use this approach.

Response 1.3: We thank the Editor and Reviewer#1 for suggesting the inclusion of a dedicated “Related Works” section to clearly situate our study within existing literature. We have now added a new section titled “Related Works”, where we discuss existing models and approaches to preeclampsia prediction, explicitly highlighting how our work differs in terms of data sources, methodological approach, scope, interpretability, and clinical integration potential. Additionally, we have included the recommended study (“ILITIA: telehealth architecture for high-risk gestation classification”) to contrast our machine learning-based approach with alternative specialized architectures.

The section now reads:

Related Works

Existing clinical approaches to preeclampsia risk prediction have significant limitations, often missing up to 66% of cases due to reliance on limited clinical risk factors, such as prior history, chronic hypertension, or diabetes alone [8, 9]. Over the past decade, multiple predictive models (Table 2) have been developed using varied data sources, ranging from basic clinical factors and vital signs to advanced biomarkers, genetics, and ultrasound findings [8-14].

Early statistical models using logistic regression (e.g., the Fetal Medicine Foundation’s first-trimester combined screening test) incorporated clinical, biochemical (Pregnancy-associated Plasma Protein-A, PAPP-A, placental growth factor, PlGF), and biophysical markers (mean arterial pressure, uterine artery Doppler) to achieve high accuracy (AUC ~0.90) for early-onset preeclampsia[10, 15]. However, these require specialized tests not routinely available, limiting broader applicability.

Biomarker-based approaches, particularly the soluble fms-like tyrosine kinase-1/placental growth factor (sFlt-1/PlGF) ratio, demonstrate strong predictive performance (high negative predictive value) in symptomatic women later in pregnancy and are used clinically in some European guidelines[16]. Still, their utility for general early-pregnancy screening remains limited due to timing and resource constraints.

More recently, machine learning methods, such as random forests, gradient boosting (xgboost), and neural networks, have shown potential by capturing complex nonlinear relationships inherent in clinical data[13, 15, 17, 18]. For example, models by Jhee et al. and Marić et al. using gradient boosting and elastic net methods achieved AUCs ranging from 0.79–0.92 across various datasets and gestational ages, typically outperforming simpler logistic models[13, 18]. Yet, deep learning methods have not consistently demonstrated superiority over ensemble models when applied to structured prenatal data[19].

Despite promising accuracy, most ML models have limitations, including insufficient external validation, reliance on non-routine biomarkers, and lack of interpretability, limiting clinical adoption[17, 19]. For instance, Li et al. employed longitudinal EHR-based machine learning models with promising internal results but lacked external validation[12]. Specialized telehealth architectures, such as ILITIA by Moreira et al., utilize rule-based systems that simplify remote screening but do not fully leverage the complex relationships captured by machine learning[20].

Table 2. Performance and description of models to predict preeclampsia risk

Study Model Type AUC Patient Population Timing Key Predictors

Wright et al. (2019) [10]

Logistic regression 0.80-0.95 61,174 First Trimester Clinical, Doppler, PlGF, PAPP-A

Sandstrom et al (2021) [11]

Shared-effects joint longitudinal models 0.73-0.87 58,899 nulliparous Longitudinal Clinical

Li et al. (2022) [12]

Xgboost 0.66-0.92 108,557 Longitudinal Clinical

Marić et al. (2020)[13]

Elastic Net, Gradient Boosting 0.79–0.89 16,370 First Trimester Clinical

Kovacheva et al. (2024) [14]

Xgboost and Polygenic Risk Score 0.74 1,125 pregnancies First & Second Trimester Genetic, Clinical

Espinoza et al. (2024) [16]

Biomarker (sFlt-1/PlGF Ratio) 0.91 543 Mid-to-Late Pregnancy sFlt-1/PlGF Ratio

Ranjbar et al. (2024) [17]

Random Forest, Xgboost, Elastic Net 0.86-0.97 Systematic Review First/Second Trimester Clinical, Biomarkers

Jhee et al. (2019)[18]

Gradient Boosting 0.92 11,006 Third Trimester Clinical

Liu et al. (2022)[19]

Neural Network, Random Forest 0.86 11,057 Early Pregnancy Clinical, Doppler, PAPP-A

Abbreviations: sFlt-1/PlGF, soluble fms-like tyrosine kinase-1/placental growth factor; PAPP-A, Pregnancy-associated Plasma Protein-A

Our study addresses these gaps by developing interpretable, longitudinal machine learning models using routinely collected clinical data from electronic health records, tested rigorously through internal and external validation. Unlike previous works, our approach uniquely explores temporal dynamics of predictors, uncovering novel indicators such as fluctuations in red blood cell counts, and utilizes Shapley values for enhanced interpretability and clinical relevance.

Ultimately, embedding such interpretable, machine-learning-driven predictive tools directly into routine obstetric care could transform the clinical management of preeclampsia, allowing personalized interventions and targeted surveillance.

4. Discussion on Public Health Applicability

o Expand the discussion on how this technology can be integrated into public health systems globally, particularly to ensure greater equity in early diagnosis and treatment.

o Address potential challenges in adopting this tool in countries with varying healthcare access and discuss possible strategies for implementation.

Response 1.4: We appreciate the Editor and Reviewer#1’s insightful comment. We have expanded our Discussion section to thoroughly address the public health implications of our predictive tool, specifically its potential integration into global public health systems, equity considerations, and strategies to overcome implementation challenges. The interpretability and reliance on routine clinical variables uniquely position our model for global implementation, ensuring accessibility and scalability even in resource-limited settings.

We added the following text to the Discussion: Our predictive modeling approach offers significant promise for integration into public health systems globally, potentially improving maternal and neonatal health outcomes through earlier identification and proactive management of preeclampsia. By relying solely on routinely collected clinical data available through standard EHR data, this model would be feasible to implement across diverse healthcare settings, including low-resource environments. However, several challenges must be addressed to facilitate broad global adoption. Limited infrastructure and inconsistent EHR availability in low- and middle-income countries could present significant barriers. To mitigate these challenges, implementation strategies could include developing simplified mobile-health (mHealth) adaptations, leveraging widely available technologies such as smartphones, and creating streamlined, low-cost data capture systems. Additionally, ensuring model transferability and robustness across diverse populations requires local validation and potential retraining of predictive models to account for population-specific risk factors and variations in clinical practice.

5. Prospective Validation

o Include a brief discussion on future strategies for prospective validation of the model and its importance in consolidating clinical applicability.

Response 1.5: We thank the reviewer for highlighting the critical need for prospective validation. We have included a brief discussion in the manuscript emphasizing strategies for prospective validation and its significance in confirming the clinical utility of our predictive models.

We added the following text to the Discussion: Prospective validation represents a critical next step to ensure the clinical applicability and utility of our predictive models. Future research should involve deploying this model prospectively in clinical practice within diverse healthcare settings, carefully assessing real-world predictive performance, clinician usability, patient outcomes, and cost-effectiveness. Ultimately, such validation is essential to establish confidence among clinicians and policymakers, driving widespread adoption and ensuring tangible improvements in maternal and neonatal health outcomes.

6. Improving Accessibility and Clarity

o Review the text to enhance its accessibility for a broader audience, adding further explanations where needed and, if applicable, providing supplementary material for technical concepts.

Response 1.6: We appreciate the reviewer’s suggestion to enhance accessibility and clarity for a broader audience. We have carefully reviewed the manuscript and revised the text, ensuring greater readability by clarifying technical terms and methodological approaches. We have also provided supplementary materials, including definitions of technical concepts, a detailed flowchart illustrating the methodological steps (Fig. 1), and additional interpretability explanations (e.g., simplified descriptions of SHAP values). These materials will ensure broader accessibility, particularly for readers who may not have a specialized background in machine learning or clinical informatics.

We also added the following text and Table 1 in the Introduction:

Given the technical complexity of machine learning terminology, we have included definitions and clinical relevance of key terms (Table 1) to facilitate reader comprehension.

Table 1. Explanation of Machine Learning and Statistical Terms

Term Definition and Clinical Relevance

Machine Learning (ML) A subset of artificial intelligence techniques that allows algorithms to learn patterns from data without explicit rules, improving predictive accuracy.

Area Under the Curve (AUC) A statistical measure of how well a prediction model distinguishes between two classes (e.g., presence or absence of preeclampsia). AUC values range from 0.5 (no predictive power) to 1.0 (perfect prediction).

Logistic Regression A statistical method used to predict the probability of an outcome based on one or more predictor variables. Suitable for binary outcomes (e.g., presence or absence of disease).

Elastic Net A regression model combining regularization techniques to manage numerous predictors, effectively reducing model complexity and improving prediction stability.

Naive Bayes A classification algorithm based on Bayes’ theorem that assumes independence between predictor variables. Despite its simplicity, it performs well with large datasets.

Random Forest A machine learning algorithm using multiple decision trees to improve predictive accuracy and control overfitting by combining individual predictions.

Xgboost (Gradient Boosting) An advanced machine learning technique using sequential decision trees, focusing on reducing prediction errors of previous trees, providing high accuracy and efficiency with structured data.

Neural Network (Deep Learning) Computational models inspired by human neural structures, capable of identifying complex, nonlinear relationships between variables through multiple layers of interconnected nodes (‘neurons’).

SHAP (Shapley) Values A technique providing interpretability of machine learning predictions by quantifying the impact of each feature on individual and overall predictions, based on cooperative game theory.

Data Leakage A methodological error where information from outside the training set is inadvertently used during model training, leading to falsely inflated performance.

Oversampling (RandomOversampler) A data balancing technique used to handle imbalanced datasets by randomly duplicating examples from the minority class, thus improving the model’s ability to detect rare conditions (e.g., preeclampsia).

Internal vs. External Validation Internal validation refers to evaluating a model’s performance on a separate subset of data from the same source used for training, while external validation tests performance on an independent dataset from a different source or population.

1. Please ensure

---

## [Editor Report · Decision Letter 1]

16 Apr 2025

Development and validation of an interpretable longitudinal preeclampsia risk prediction using machine learning

PONE-D-24-59081R1

Dear Dr. Kovacheva,

We’re pleased to inform you that your manuscript has been judged scientifically suitable for publication and will be formally accepted for publication once it meets all outstanding technical requirements.

Kind regards,

Zenewton André da Silva Gama, Ph.D.

Academic Editor

PLOS ONE

---

## [Editor Report · Acceptance letter]

PONE-D-24-59081R1

PLOS ONE

Dear Dr. Kovacheva,

I'm pleased to inform you that your manuscript has been deemed suitable for publication in PLOS ONE. Congratulations! Your manuscript is now being handed over to our production team.

Kind regards,

on behalf of

Prof. Dr. Zenewton André da Silva Gama

Academic Editor

PLOS ONE